# SPHINX: VISUAL PERCEPTION AND REASONING GYM

## ABSTRACT

We present SPHINX, a synthetic gym for visual perception and reasoning tasks that targets core cognitive primitives. SPHINX procedurally generates problems using motifs, tiles, charts, icons, and geometric primitives, each paired with verifiable ground-truth solutions. This design enables both precise evaluation and the creation of scalable datasets. We implement 25 task types spanning symmetry detection, geometric transformation, spatial reasoning, chart interpretation, and sequence prediction. Benchmarking recent multimodal vision–language models (vLLMs) reveals that even state-of-the-art GPT-5 struggles on these tasks, achieving 47.32% accuracy and performing significantly below human baselines. Finally, we demonstrate that reinforcement learning with verifiable rewards (RLVR) improves model accuracy on these reasoning tasks, underscoring its potential for advancing multimodal reasoning.

## 1 INTRODUCTION

Large language models (LLMs) have recently demonstrated striking advances in reasoning, achieving gold medal level performance at the International Mathematical Olympiad Castelvecchi (2025) and strong results across mathematics, logical reasoning, and coding Guo et al. (2025); Jaech et al. (2024); Wu et al. (2024); Comanici et al. (2025); Yang et al. (2025a). Because reasoning is a core component of human intelligence, it has become a central benchmark for progress toward Artificial General Intelligence (AGI) Goertzel (2014). Techniques such as Chain-of-Thought prompting Wei et al. (2022), test-time compute scaling Jaech et al. (2024), and post-training strategies like rule-based reinforcement learning in DeepSeek-R1 have further improved model performance, helping mitigate reward hacking Guo et al. (2025) and enabling more robust generalization across domains Xie et al. (2025); Albalak et al. (2025); He et al. (2025).

In contrast to the rapid progress of LLMs, multimodal large language models (MLLMs) remain far less capable in visual reasoning. Unlike text-based systems that can leverage structured prompts and post-training strategies, MLLMs must jointly parse visual inputs and integrate them with language, a substantially more complex challenge Gandhi et al. (2025); Guo et al. (2025); Xie et al. (2025); Wang et al. (2025c). Current models often fail to construct coherent reasoning chains and stumble on tasks trivial for humans Yang et al. (2025b). While reinforcement learning has been applied to strengthen MLLMs Liu et al. (2025a); Peng et al. (2025), progress is constrained by benchmarks that emphasize perception over reasoning, such as referring expression comprehension or math-with-diagram datasets, where models frequently reduce visual inputs to text and rely on language reasoning Xu et al. (2025b); Zhang et al. (2024).

More recently, several works have begun to investigate abstract visual reasoning (AVR) in MLLMs Xu et al. (2025b); Cao et al. (2024); Małkiński et al. (2024); Jiang et al. (2024a); Lee et al. (2024); Chollet et al. (2025). Yet these efforts still fall short of systematically evaluating core perceptual primitives such as symmetry detection, mental rotation, and structured pattern matching. Cognitive science has long shown that such abilities underpin fluid intelligence and matrix reasoning Fisher et al.

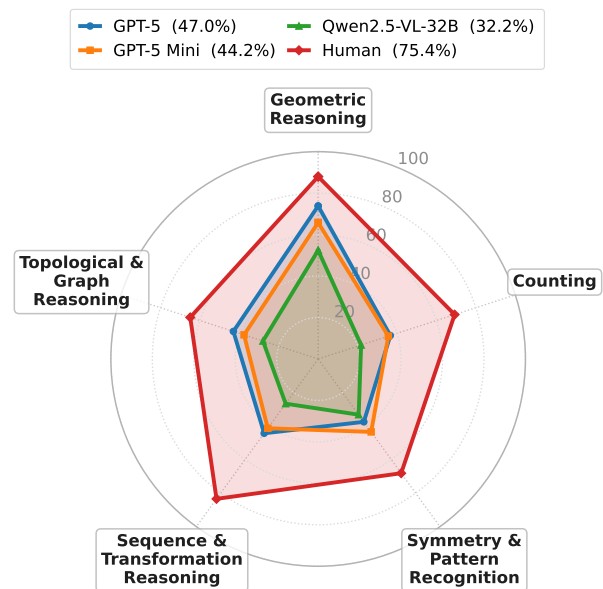

Figure 1: Radar plot shows accuracies (%) achieved by LLMs and by human on the broad categories of SPHINX.

(1981); Carpenter et al. (1990); Pizlo & De Barros (2021); Shepard & Cooper (1986). For machine learning, this suggests that practical evaluation must directly target these primitives through controlled tasks that disentangle perception from abstraction.

**SPHINX: Visual Perception and Reasoning Gym.** We introduce SPHINX, a synthetic environment for generating families of visual perception and reasoning tasks centered on symmetry, transformation, and related primitives. Each instance is paired with an unambiguous ground-truth solution, enabling precise evaluation. SPHINX serves a dual purpose: it provides controllable task generation that systematically targets different perceptual and reasoning abilities, and it offers insight into model failure modes. Moreover, the synthetic generation pipeline scales to produce datasets large enough for reinforcement learning, paralleling the role of synthetic reasoning environments in advancing text-based LLM reasoning Stojanovski et al. (2025a); Chen et al. (2025a).

**Contributions.** We make the following key contributions:

1. We introduce SPHINX, a synthetic environment for generating datasets in visual perception and reasoning, comprising 25 tasks across five broad categories (see Figure 1). To the best of our knowledge, this represents the largest-scale synthetic environment designed for such tasks.

2. We construct a benchmark dataset with 2,500 questions using SPHINX and evaluate a range of proprietary and open-source MLLMs. We provide a comparative analysis between human performance and MLLM performance across task categories.

3. We conduct reinforcement learning with verifiable rewards (RLVR) on a separate training set derived from SPHINX, demonstrating both improved in-distribution performance and the potential to generalize to out-of-distribution tasks.

## 2 SPHINX DESIGN

SPHINX is a modular framework for programmatically generating visual reasoning data with verifiable ground truth. Its central idea is to decouple appearance from rule structure through three composable modules: *motifs*, *tilings*, and *tasks*, allowing each dimension to be flexibly combined or independently varied.

### 2.1 DESIGN PRINCIPLES

1. **Factorized control of variation.** Appearance (*motifs*), spatial layout (*tilings*), and reasoning rules (*tasks*) are separated, enabling systematic exploration across perceptual diversity, geometric structures, and rule families.

2. **Verifiable supervision.** Each instance is paired with a deterministic checker that certifies rule satisfaction and guarantees a single correct answer; this eliminates ambiguity and supports exact evaluation as well as reinforcement learning with verifiable rewards (RLVR).

3. **Distribution and difficulty control.** Weighted samplers govern the mix of tasks and motifs, while difficulty knobs (e.g., step sizes, noise ranges, path lengths) provide fine-grained control over problem complexity.

4. **Standardized artifacts.** Every sample exports a composite image, natural-language prompt, ground-truth answer, distractors (if any), and rich metadata (including construction parameters and lightweight complexity scores) in analysis-ready formats.

### 2.2 BUILDING BLOCKS

**Motifs (rendered primitives).** A motif is a parameterized renderer $m(\theta)$ that produces an RGBA tile from attributes such as kind, size, count, angle, and stroke. Families include dots, rings, polygons and star polygons, crescents, glyphs, and other iconographic primitives. Motifs expose attribute ranges and a *clamp* to guarantee validity; tasks can bias selection via per-task motif weights and request asymmetric variants to avoid trivial self-mappings in symmetry/transform problems. Example motifs are shown in Figure 2.

**Geometric primitives.** Beyond motifs, SPHINX renders canonical geometry shapes including circles, n-gons, angles, polylines constrained to grid edges, grids, and Venn-style region unions. These support tasks hinge on spatial relations and combinatorial structure (e.g., symmetry classification, shortest paths, connected components, region area/perimeter).

**Tilings (geometric canvases).** Tilings define cell layouts and adjacency (square, triangular, hexagonal, rhombille, and circle variants). Tiling specs control grid size, margins, adjacency notion, and coloring regime. Uniform schemes and palette-driven non-uniform schemes yield structured variation. Example tilings are shown in Figure 3.

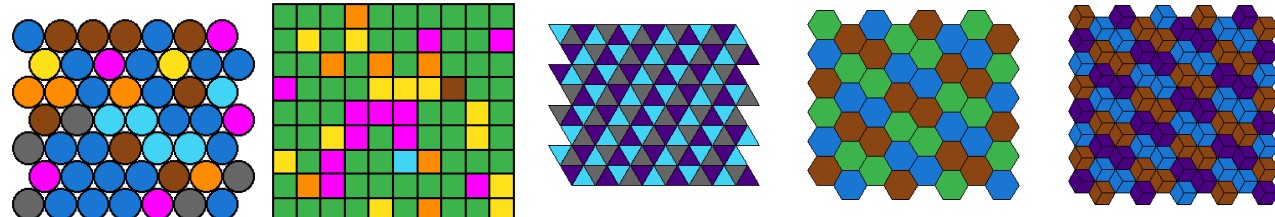

Figure 2: Example Motifs (from left): Crescent, Glyph, Pinwheel, Polygon, Polyomino and Icons

Figure 3: Example Tilings: circles, square, triangular, hexagonal, rhombille.

## 2.3 TASKS

A task $r$ maps one or more motif instances and/or tiled regions into a well-defined question, optionally with multiple-choice options. Each instance outputs a composite image, a natural-language prompt, and precisely one unique correct answer, along with distractors when applicable. For MCQ formats, all options are rendered with consistent borders and labels to eliminate formatting cues. A key design principle in our task formulation is that questions should be visually answerable directly from the image by a human, without requiring detailed, paper-and-pencil style reasoning.

We categorize the tasks into five broad families. Figure 4 illustrates representative examples, with additional cases provided in the Appendix.

**Geometric Reasoning.** This category comprises tasks where spatial relations, shape sizes, areas, perimeters, or comparative geometry are the key factors. Such problems align with relational and geometric reasoning in the literature, focusing on spatial arrangements and geometric properties, and with formal geometric reasoning tasks that require constructing and analyzing geometric diagrams Lu et al. (2021a); Zhang et al. (2024). The tasks include:

1. **Positional Count**: Count how many small shapes satisfy a specific spatial relation (inside, outside, above, below) relative to larger reference shapes.

2. **Shape Sorting**): Sort a set of geometric shapes (polygons, ellipses, angles, lines) by area, perimeter, or angle measure.

3. **Stack Count**: Count objects that lie strictly inside a specified sheet in a stack of overlapping shapes, where only the top shapes are fully visible.

4. **Pie Chart**: Rank the slices of a pie chart by their visual size.

5. **Chart Comparison**: Match a pie chart with a bar chart by visually comparing the relative proportions of their segments.

**Counting.** Tasks in this group focus on counting discrete elements or measuring linear features in visual scenes, akin to the counting and comparison tasks emphasised by early diagnostic benchmarks such as CLEVR Johnson et al. (2017). They include:

6. **Venn Diagram**: Compute sums in different regions of a Venn diagram rendered with overlapping shapes.

7. **Shape Counting**: Count the number of sub-shapes (e.g., rectangles, squares, triangles, parallelograms) contained within a composite figure.

8. **Tiles Line Length**: Measure the length of a highlighted polyline in a tiling by counting edge steps.

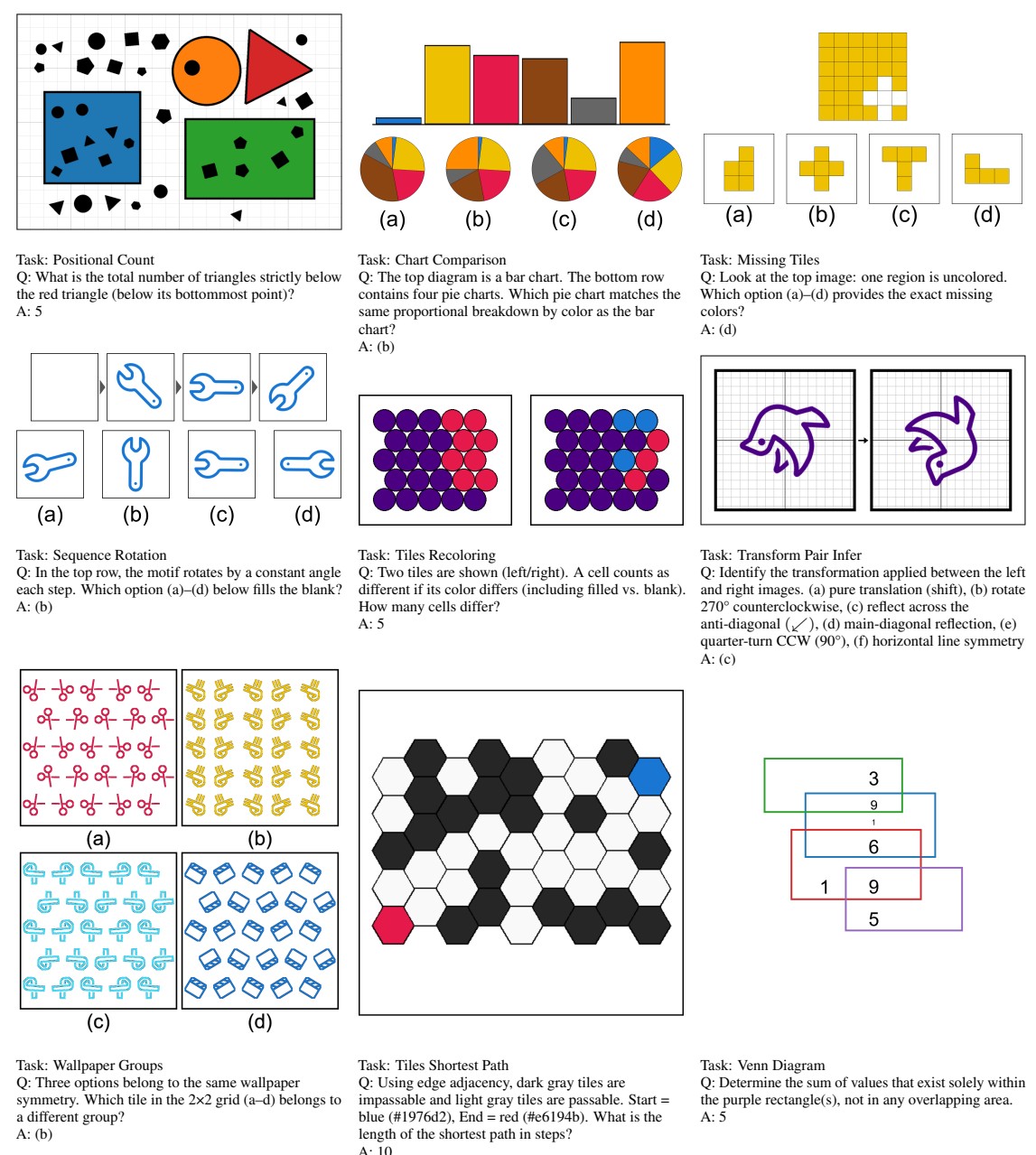

Figure 4: SPHINX task illustrations

9. **Tiles Line Intersections**: Count the intersection points between coloured polylines constrained to tile edges.

10. **Tiles Recoloring**: Count the number of cells that differ between two coloured boards, typically reflecting the size of a modified region.

**Symmetry & Pattern Recognition.** These tasks require detecting symmetry, periodicity, or odd-one-out patterns. Similar phenomena are explored in visual oddity and abstract reasoning benchmarks, where participants must identify the element that violates a geometric rule or pattern Zerroug et al. (2022b); Woźniak et al. (2023). The SPHINX tasks are:

11. **Mirror Identification**: Classify an image according to the type of mirror symmetry present.

12. **Symmetry Fill**: Complete a $2 \times 2$ grid by selecting the tile that satisfies a specified mirror symmetry.

| Models | Overall | Geometric Reasoning | Counting | Symmetry & Pattern Recognition | Sequence & Transformation | Topological & Graph Reasoning |
|---|---|---|---|---|---|---|
| **Reference** | | | | | | |
| Human | 75.39 | 87.97 | 69.23 | 68.14 | 83.43 | 64.89 |
| **Closed Source LLMs** | | | | | | |
| GPT-5 | 47.32 | 73.80 | 36.60 | 37.50 | 44.33 | 43.00 |
| GPT-5 Mini | 44.68 | 65.80 | 35.60 | 43.50 | 41.33 | 37.60 |
| GPT-5 Nano | 32.44 | 44.40 | 24.60 | 39.25 | 31.50 | 24.00 |
| **Open Source LLMs** | | | | | | |
| InternVL3-8B | 18.28 | 27.60 | 13.00 | 18.25 | 16.00 | 17.00 |
| InternVL3-38B | 25.08 | 41.00 | 18.80 | 18.25 | 23.00 | 23.40 |
| Llama-3.2-11B | 14.64 | 17.40 | 1.60 | 20.00 | 23.67 | 9.80 |
| Qwen2.5-VL-3B | 16.96 | 27.80 | 7.00 | 16.75 | 19.83 | 12.80 |
| Qwen2.5-VL-7B | 24.08 | 37.80 | 14.60 | 28.25 | 22.83 | 18.00 |
| Qwen2.5-VL-32B | 32.16 | 52.40 | 21.80 | 33.25 | 26.67 | 28.00 |

Table 1: Performance comparison of human, close-source, and open-source LLMs across multiple reasoning categories.

13. **Frieze Groups**: In a set of four frieze patterns, identify the one that belongs to a different symmetry group.

14. **Wallpaper Groups**: Identify the odd patch among four wallpaper patterns.

**Sequence & Transformation Reasoning.**   This category encompasses tasks involving temporal sequences, rotation progressions, or transformation inference. These tasks correspond to temporal reasoning and mental-rotation challenges Wexler et al. (1998). The tasks include:

15. **Transform Result Identify**: Choose the correct result when a specific transformation is applied to an image.

16. **Transform Pair Infer**: Given two tiles, determine the transformation that maps the source to the target.

17. **Transform Similarity Identify**: Identify which option is similar or dissimilar to a base shape under Euclidean similarity transformations (uniform scaling, rotation, reflection).

18. **Sequence Rotation**: Predict the missing panel in a sequence of rotated motifs.

19. **Sequence Arithmetic**: Predict the missing panel in a numeric progression of shapes.

20. **Sequence Multi-Column Arithmetic**: Predict the next panel when each column in a grid independently undergoes its own arithmetic progression.

**Topological & Graph Reasoning.**   These tasks involve reasoning about connectivity, paths, and assembly on tilings or grids. Graph-reasoning benchmarks classify such problems under path-query and connectivity tasks Wei et al. (2024). The tasks are:

21. **Tiles Geometry**: Compute areas, perimeters, number of holes, or union perimeters of colored regions on a tiling.

22. **Tiles Connected Component**: Determine the size or number of connected components of a specified colour under different adjacency notions.

23. **Tiles Shortest Path**: Find the minimal number of steps between two tiles or determine that no path exists.

24. **Missing Tiles**: Restore missing tiles by selecting shapes or colour assignments that fit the blanked region.

25. **Tiles Composition**: Decompose a connected region into smaller pieces or compose multiple pieces into a single connected shape.

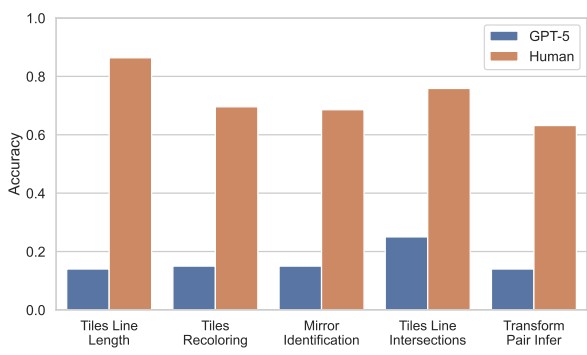

Figure 6: Tasks where humans exceed GPT-5.

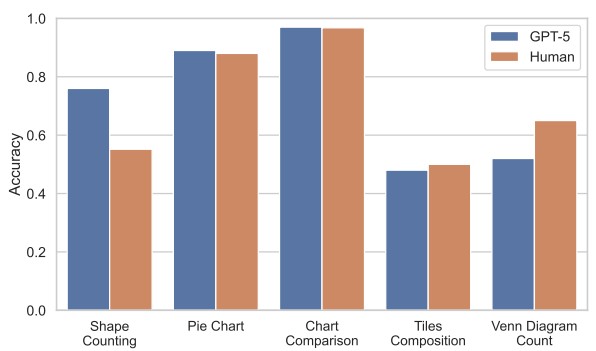

Figure 7: Tasks where GPT-5 exceeds or is close to human performance.

## 3 BENCHMARK

We curated the SPHINX benchmark to consist of 2,500 questions, with 100 instances per task. We evaluate three proprietary ChatGPT-5 variants (regular, mini, and nano) using their default reasoning settings OpenAI (2025). In addition, we assess six open-source vision–language models (VLMs), including the Qwen2.5 family Bai et al. (2025), Llama 3.2 Meta (2024), and InternVL3 Zhu et al. (2025), spanning parameter scales from 3B to 38B.

The results are summarized in Table 1. Overall, GPT-5 achieves the best performance with an average accuracy of 47.32% across tasks, although it still falls short of human accuracy by 28.07%. GPT-5 mini performs comparably, with only a 2.64% drop relative to the regular model. Among open-source models, Qwen2.5-VL-32B achieves the highest accuracy (32.16%), followed by InternVL3-38B at 25.08%.

Performance varies substantially across task categories. The most significant gap between models and human evaluators occurs in *Sequence* and *Transformation* tasks, where GPT-5 lags human accuracy by 39.2%. In contrast, the gap is less pronounced on *Geometric Reasoning* and *Tiles*-based tasks that emphasize topological or graph-structured reasoning. Figure 5 shows performance across all 25 tasks, comparing GPT-5 with human accuracy. While there is an overall positive correlation, several tasks exhibit substantial disparities, which we analyze in detail below.

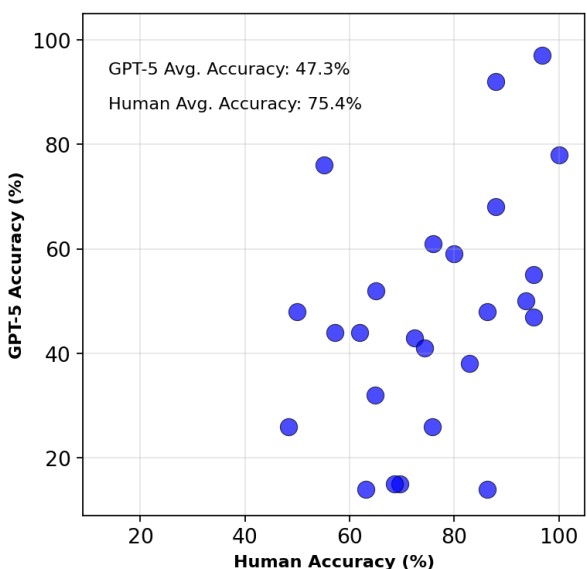

Figure 5: Comparison of human and GPT-5 accuracy.

## 4 ANALYSIS

### 4.1 GPT-5 VS. HUMANS

In Figure 6, we present the five tasks where human performance most clearly surpasses GPT-5, while Figure 7 highlights the opposite cases where GPT-5 performs comparably to or better than humans. Three of the tasks where GPT-5 struggles involve reasoning over tiles (*Tiles Line Length*, *Tiles Recoloring*, and *Tiles Line Intersections*), which humans find substantially easier. The remaining two tasks involve identifying mirror symmetry and inferring transformations between paired images. Figure 10(left) shows an example of GPT-5 incorrect response for the *Tiles Line Length* task.

Conversely, GPT-5 fares much better on tasks involving counting over plain backgrounds with geometric shapes, where humans may struggle due to the shapes being relatively small compared to the overall image.

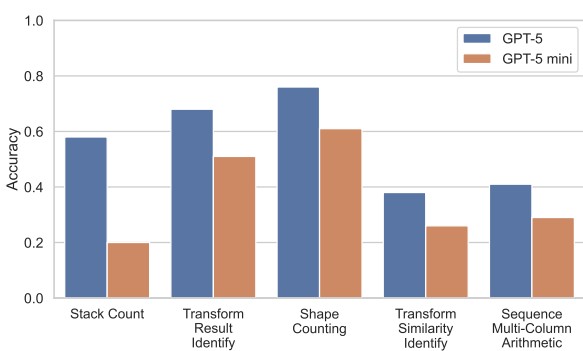

Figure 8: Tasks where GPT-5 exceeds GPT-5 mini

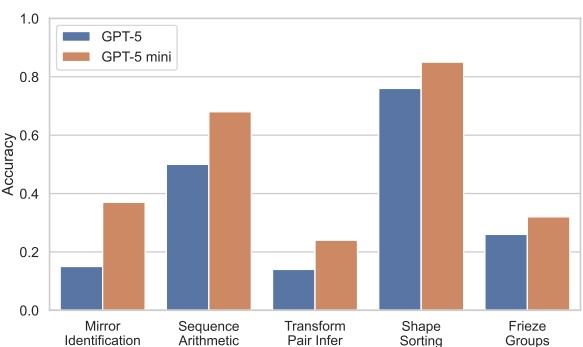

Figure 9: Tasks where GPT-5 mini exceeds GPT-5.

Additional tasks where GPT-5 approaches human performance include *Pie Chart* and *Chart Comparison* from geometric reasoning, as well as *Tiles Composition* from topological and graph reasoning. Notably, these are also among the tasks where human evaluators performed the worst across all 25 tasks.

## 4.2 GPT-5 VS. GPT-5 MINI

(Figure 8 and Figure 9) compare the five tasks where GPT-5 outperforms GPT-5 mini and vice versa, with one representative pair of example responses shown in Figure 10 (middle and right). We find that GPT-5 generally performs better on tasks with explicit instructions, such as counting or identifying the result of a specified transformation. In contrast, GPT-5 mini performs better on tasks that require no explicit guidance, where the model must infer underlying rules to answer correctly, such as symmetry identification or transformation inference. This contrast highlights the tendency of larger MLLMs to "overthink" certain problems, whereas smaller variants may benefit from relying on simpler heuristics.

## 5 REINFORCEMENT LEARNING WITH VERIFIABLE REWARD

We perform reinforcement learning with verifiable rewards on synthetic datasets generated by SPHINX.

**Data Split.** We designate 20 tasks as in-distribution and withhold five tasks from training to assess generalization to unseen tasks. The withheld tasks are *Geometric Position Count*, *Tiles Recoloring*, *Wallpaper Groups*, *Sequence Multi-Column Arithmetic*, and *Tiles Composition*. We generate $100,000$ synthetic samples using a fixed random seed. From these, we select $1,600$ samples per in-distribution task (a total of $32,000$ training samples) such that the *minimum* semantic similarity (with respect to evaluation samples of the same task) is maximized. Semantic similarity is computed using the `sentence-transformers` library Reimers & Gurevych (2019), employing the CLIP ViT-B/32 embedding model.

**Model Training.** We train using GRPO (Group Relative Policy Optimization), an RL method that eliminates the need for a separate value (critic) network by ranking multiple outputs per prompt and using their relative scores as a baseline Shao et al. (2024). Our base model is Qwen2.5-7B and 3B parameter Bai et al. (2025), fine-tuned using the EasyR1 framework Yaowei Zheng (2025). Training is conducted for 100 iterations with hyperparameters set as follows: kl_coef $= 1.0 \times 10^{-2}$, maximum response length = 2048, optimizer = `adamw` (learning rate $1.0 \times 10^{-6}$, weight decay $1.0 \times 10^{-2}$), rollout parameters $n = 5$, temperature = 1.0, batch size = 128, and total 500 training steps.

We use the default prompt and reward from EasyR1 framework Yaowei Zheng (2025). We use a binary correctness reward: if a generated response is correct, it receives a reward of 1; otherwise, it receives a reward of 0. We also include a format reward, combining the two as

$$\text{reward} = \lambda \cdot \text{format\_reward} + (1 - \lambda) \cdot \text{correctness\_reward},$$

with $\lambda = 0.1$. Correctness is verified against the ground truth using the `mathruler` library hiyouga (2025).

**Model Performance.** Table 2 reports the performance of RLVR-trained models across different datasets. Along with the IID and OOD splits of the SPHINX benchmark, we also evaluate on three external benchmarks: MathVision Wang

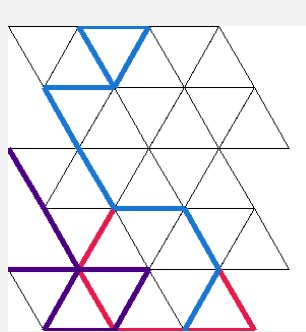

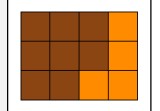 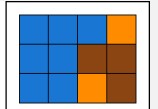

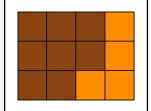 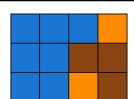

**Question:** Measure the blue (#1976d2) line strictly along tile edges. How many tile-edge units long is it?

**Response from GPT-5:**
To measure the length, count the number of unit edges... The blue path uses 4 units in each of the three directions, so the total is $4 + 4 + 4 = 12$.
| 12 |

**Correct output:** 9

**Question:** Consider color differences only (presence/absence included). How many cells differ between the tiles?

**Response from GPT-5:**
Step-by-step:
- Represent each tile as a 3×4 grid...
- Left tile colors:
- r1: [brown, brown, brown, orange]... - Right tile colors:
- r1: [blue, blue, blue, orange]... - Compare cell by cell. Matching cells: (r1,c4), (r2,c3), (r3,c3) → 3 matches.
- Total cells = 12, so differing cells = 12 - 3 = 9.
| 9 |

**Correct output:** 9

**Question:** Consider color differences only (presence/absence included). How many cells differ between the tiles?

**Response from GPT-5 mini:**
Label rows 1–3... and columns 1–4... Reading the two tiles:
Left tile:
- Row1: brown, brown, brown, orange...
Right tile:
- Row1: blue, blue, blue, orange...
Compare each... cell; cells that change color are: (1,1),(1,2),(1,3),(2,1), (2,2),(3,1),(3,2),(3,4) — 8 cells.
Answer: | 8 |

**Correct output:** 9

(a) GPT-5 incorrect response for *Tiles Line Length*

(b) GPT-5 response correct response for *Tiles Recoloring*.

(c) GPT-5 mini incorrect response for *Tiles Recoloring*.

Figure 10: Three qualitative examples of model responses on visual reasoning tasks.

Table 2: Performance of Qwen2.5 models with and without RLVR across benchmarks. Values are accuracies (%).

| Model | Sphinx IID | Sphinx OOD | MathVision | MM-IQ | Geo3k |
|---|---|---|---|---|---|
| Qwen2.5-7B | 25.15 | 19.8 | 16.8 | 24.6 | 37.9 |
| +RLVR | 42.55 | 26.8 | 23.9 | 25.9 | 37.1 |
| Qwen2.5-3B | 17.55 | 14.6 | 21.8 | 22.8 | 24.5 |
| +RLVR | 31.65 | 22.2 | 21.8 | 24.7 | 29.0 |

et al. (2024), MM-IQ Cai et al. (2025), and Geo3k Lu et al. (2021a). We use the same prompting and evaluation setup as in training for all datasets.

We observe substantial performance gains on the IID split of the 20 shared tasks between training and testing for both models, and these improvements also transfer to the five OOD tasks, with Qwen2.5-7B improving by 7%. Results on the three external datasets are more mixed: we see improvements in some cases, such as MathVision for the 7B model and Geo3k for the 3B model, but gains are not consistent across benchmarks. We hypothesize that closer integration of our synthetic datasets with existing benchmarks could yield more systematic improvements.

## 6 RELATED WORKS

Research on visual reasoning has long been motivated by studies in psychology and cognitive science. Human cognition is often assessed through tests such as Raven's Progressive Matrices (RPM) Carpenter et al. (1990) and the Wechsler Intelligence Scale for Children (WISC) Wechsler (1949), which measure abstraction, analogy, and fluid intelligence. These tasks emphasize core perceptual and reasoning primitives, such as symmetry detection, pattern completion, and spatial transformation, that emerge early in human development and remain challenging for artificial systems.

**Datasets and fixed benchmarks.** Inspired by these traditions, many datasets adapt cognitive test formats for evaluating models. The Abstraction and Reasoning Corpus (ARC) Lee et al. (2024), Bongard Problems Małkiński et al. (2025), and BONGARD-LOGO Nie et al. (2020) probe concept learning and analogy-making. IQ-inspired datasets such as MM-IQ Cai et al. (2025), MARVEL Jiang et al. (2024b), and SMART-101 Cherian et al. (2023) measure abstraction and generalization using puzzles originally designed for standardized exams or children's competitions. MATH-Vision Wang et al. (2024) targets multimodal mathematical reasoning, while MaRs-VQA Cao et al. (2025) provides psychologist-certified matrix reasoning tests to compare humans and multimodal models. Reviews of RPM-solving methods Małkiński & Mańdziuk (2025b) consistently highlight large human–model performance gaps, particularly in zero-shot generalization. While these datasets reveal important weaknesses, they are typically fixed in size and limited in diversity.

**Synthetic and procedural benchmarks.** To overcome the limitations of fixed datasets, several works adopt procedural generation. Compositional Visual Reasoning (CVR) Zerroug et al. (2022a), A-I-RAVEN and I-RAVEN-Mesh Małkiński & Mańdziuk (2025a), and NTSEBench Pandya et al. (2025) extend RPM-like designs with controlled variation. IconQA Lu et al. (2021b) introduces programmatically generated diagram problems, while VisuLogic Xu et al. (2025a) and Visual Riddles Bitton-Guetta et al. (2024) emphasize multimodal abstraction and commonsense puzzles. Broader synthetic environments include Reasoning Gym Stojanovski et al. (2025b), Enigmata Chen et al. (2025b), and UniBench Al-Tahan et al. (2024), which demonstrate scalable generator–verifier frameworks or unified evaluation protocols. Despite these advances, most efforts focus on narrow domains or lack integrated verifiable feedback. SPHINX builds on this line of work by offering procedurally generated problems that span a wide range of perceptual and reasoning categories, each paired with deterministic verifiers for precise and repeatable evaluation.

**Reinforcement learning for visual reasoning.** Recent work has explored reinforcement learning with verifiable rewards (RLVR) to improve model reasoning. Reason-RFT Tan et al. (2025), Visual-RFT Liu et al. (2025b), and Jigsaw-R1 Wang et al. (2025b) demonstrate that reinforcement fine-tuning improves generalization in visual reasoning tasks. ViGoRL Sarch et al. (2025) grounds reasoning steps spatially for interpretability, while MoDoMoDo Liang et al. (2025) investigates data mixture strategies. VL-Rethinker Wang et al. (2025a) and VL-Cogito Yuan et al. (2025) further incorporate RL for self-reflection and curriculum-based training. Generator–verifier setups such as Reasoning Gym Stojanovski et al. (2025b) and Enigmata Chen et al. (2025b) underscore the importance of scalable reward signals. SPHINX complements these approaches by providing a synthetic gym where every task has a verifiable ground-truth solution, making it naturally suited for RLVR experiments.

## 7 LIMITATIONS & FUTURE WORK

While SPHINX provides a large-scale synthetic environment for visual perception and reasoning, our current focus is limited to a specific set of task families. As a result, performance gains may not fully translate to more diverse benchmarks. Future work should expand the range of task types to capture the breadth of multimodal reasoning challenges better. Additionally, curriculum training strategies that explicitly incorporate task difficulty could further enhance model generalization Stojanovski et al. (2025b). Another important direction is reducing the guessability of multiple-choice questions during RL training, ensuring that improvements arise from genuine reasoning rather than shortcut exploitation Guo et al. (2025).

## 8 CONCLUSION

We introduced SPHINX, a synthetic gym for visual perception and reasoning tasks. It currently implements twelve tasks, and our evaluation shows that state-of-the-art multimodal LLMs struggle on most of them, while reinforcement learning with verifiable rewards (RLVR) offers promising gains. Future work will expand SPHINX with additional tasks in visual puzzles, geometric and spatial reasoning, and multi-step transformations, alongside improved reinforcement learning paradigms. We plan to release the framework as open source to support broader adoption and community extensions.

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

APPENDIX

# A  IMPLEMENTATION SUMMARY

## A.1  OVERVIEW

SPHINX is a framework for generating visual reasoning tasks by pairing a registry of motifs or tiles with a registry of task classes. Each task produces an instance consisting of a composed image, the specifications of its component motifs, and task metadata. Tasks are discovered dynamically and sampled according to configurable weights, enabling controlled variation during dataset generation. Some tasks include visual multiple-choice options; in these cases, distractors are constructed to be unique and clearly distinct from the ground-truth answer. Other tasks have text-based multiple-choice formats or integer outputs. To further increase variety, we use ten prompt templates for each task. The engine selects a task, samples motif specifications, renders the composite scene, and records metadata such as the question, answer, and distractors.

## A.2  TASK SUMMARIES

SPHINX currently implements twelve tasks grouped into four categories: **symmetry**, **sequence**, **tiles**, and **transformation**.

**Symmetry.**

- *Symmetry grid mirror fill*: Generates a $2 \times 2$ grid with one blank cell; the solver must choose the option that completes the grid according to a specified mirror symmetry (vertical, horizontal, or diagonal).
- *Symmetry scene mirror identify*: Arranges motifs on a canvas according to a sampled mirror symmetry and asks the model to classify whether the scene exhibits vertical, horizontal, diagonal, or no symmetry.
- *Symmetry wallpaper groups*: Presents four tiling patches from wallpaper-group symmetries, three of which share the same class while one differs; the solver must detect the odd one out.

**Sequence.**

- *Sequence arithmetic*: Shows a row of motifs whose counts follow an arithmetic progression, with one panel masked; the solver selects the missing panel from candidate options. In half the prompts, the arithmetic rule is explicitly stated, while in the other half the task is posed more generally without hints.
- *Sequence rotation*: Displays motifs undergoing a constant rotational step across panels, with one rotation hidden; the solver identifies the correct missing rotation.
- *Sequence multi-column arithmetic*: Extends the arithmetic progression task to a grid where each column evolves independently; the solver must recover the missing entry from visual options.

**Tiles.**

- *Tiles connected component*: Requires counting connected components of a given color or identifying the largest or smallest connected region.
- *Tiles shortest path*: Presents start and end cells on a tiling with obstacles, and the solver must compute the minimal-step path under adjacency constraints.
- *Tiles missing tiles*: Shows a partially occluded tiling, and the solver selects the missing piece that completes it, with rotations or reflections allowed.
- *Tiles geometry*: Asks questions about geometric properties of regions, such as area, perimeter, or the number of enclosed voids.

**Transformation.**

- *Transform result identify*: Shows a source tile and a specified transformation; the solver must select the correctly transformed result from candidate options.
- *Transform pair infer*: Presents a source and target tile and asks the solver to identify the transformation (rotation, reflection, transposition, or none) that maps one to the other.

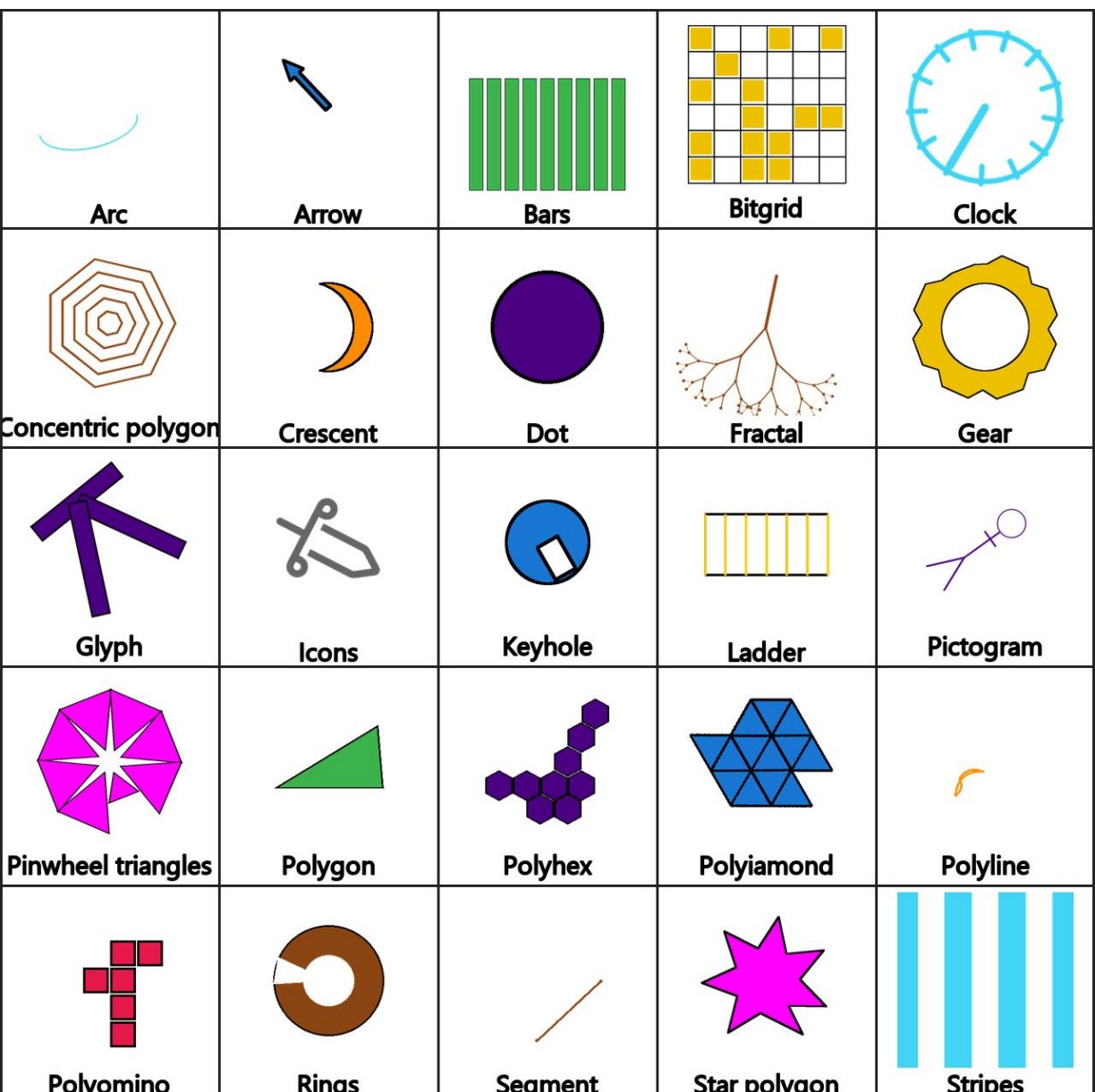

Figure 11: Randomly sampled example from each Motif family.

### A.3 IMPLEMNTED MOTIFS

Figure 11 shows representative instances of the 25 motifs currently implemented in SPHINX.

## B TASK DESCRIPTIONS

### B.1 GEOMETRIC REASONING

Figure 12 shows examples of the this type of task.

### B.1.1 POSITIONAL COUNT

**Problem.** Positional counting relative to non-overlapping reference shapes (rectangles, circles, triangles). The objective is to count small shapes satisfying a strict spatial relation to a chosen reference.

**Construction.** Place 1-4 large reference shapes with enough separation. Sample small shapes (circle, triangle, square, pentagon, hexagon) with pairwise non-overlap and strict clearance from all reference boundaries. Evaluate strict, radius-aware predicates (inside, outside, above, below, left, right) to form the label.

**Variants.** Six relation categories crossed with multiple small-shape kinds; background and counts vary with seed.

**Difficulty controls.** We measure difficulty with the count of correct shapes.

**Answer type.** Integer count.

### B.1.2 SHAPE SORTING

**Problem.** Ordinal sorting over labeled geometric primitives under a specified metric.

**Construction.** Sample a family (polygon, ellipse, angle, line) and a metric (polygon/ellipse area or perimeter; angle measure; line length). Sample values with a minimum relative gap and render using a random-pack layout with uniform label font height.

**Variants.** Four families with metrics as above; the number of items $k$ is drawn from configured bounds.

**Difficulty controls.** We control the difficulty with the number of items $k$.

**Answer type.** Free-form ranking over letters (comma-separated). No explicit distractors.

### B.1.3 STACK COUNT

**Problem.** Given overlapping sheets of equal area, count small objects that lie strictly inside a designated non-top sheet.

**Construction.** Choose a stack kind (rectangle, circle, equilateral triangle). Generate $k$ sheets with controlled pairwise overlap ratios and identical area; draw small objects (circle, triangle, square) on top of the stack. Pose an inside-of-border query about an occluded sheet.

**Variants.** Three stack families $\times$ three small-object kinds. Prompts vary in target sheet (color) and object kind.

**Difficulty controls.** We control difficulty with the number of correct shapes.

**Answer type.** Integer count.

### B.1.4 PIE CHART

**Problem.** Ordinal reasoning over a single pie chart. The model must rank categories by slice size (ascending or descending) without access to numeric labels.

**Construction.** Sample $k$ categories with percentages satisfying a strict relative gap; optionally derive consistent integer counts for provenance. Render a legend-only chart (values hidden in the pie).

**Variants.** Four light variants induced by the crossing of sort direction (ascending/descending, 50/50) and value kind (percentage vs. count), with $k$ spanning the configured range.

**Difficulty controls.** The number of $k$ categories controls the difficulty of the problem.

**Answer type.** Free-form categorical ranking (letters only, comma-separated). No multiple-choice distractors are presented.

### B.1.5 CHART COMPARISON

**Problem.** Proportion matching across two charts. A top chart (pie or bar) defines the color$\rightarrow$percentage mapping; the choice set comprises four options of the opposite chart type. Exactly one option preserves the mapping.

**Construction.** Sample $k$ categories, distinct integer percentages for the categories that sum to 100, and a distinct color palette.

**Variants.** Two display regimes with the top chart as a pie chart or a bar chart and the options as the opposite chart type.

**Difficulty controls.** We control difficulty by adjusting how many $k$ categories are in the charts.

**Distractors.** Wrong options are produced by jittering and/or permuting the percentage vector. Candidates are admitted only if they pass absolute/relative difference thresholds and pairwise image-level distinctness checks.

## B.2 COUNTING

Figure 13 shows examples of the this type of task.

### B.2.1 VENN DIAGRAM

**Problem.** Inclusion/exclusion over axis-aligned shapes with per-region numeric labels.

**Construction.** Sample 2-4 axis-aligned rectangles or ellipses with a connected union. Induce a partition grid, place one integer in each non-empty atomic region (with skinny-region fallbacks), and pose include/exclude queries whose truth set uniquely determines the sum.

**Variants.** Two layout families (rectangles vs. ellipses) with 2-4 sets.

**Difficulty controls.** We control difficulty with the number of atomic regions.

**Answer type.** Integer sum.

### B.2.2 SHAPE COUNTING

**Problem.** Counting of sub-shapes (rectangles, squares, triangles, parallelograms) within a single connected figure.

**Construction.** Draw one connected figure using one of several generators (axis-aligned polyomino, skewed poly-parallelogram, irregular/regular grids, staircase, triangular lattice, inscribed overlay). Render on a plain white background and compute the ground-truth count using exact combinatorial routines matched to the generator.

**Variants.** Eleven generator families (as above), each paired with appropriate query types. Instances are only emitted when the computed answer lies within configured bounds.

**Difficulty controls.** The number of shapes in a figure.

**Answer type.** Integer count; no multiple-choice choice set.

### B.2.3 TILES LINE LENGTH

**Problem.** Edge-step length estimation for a highlighted colored polyline.

**Construction.** On a chosen tiling, sample $k$ non-overlapping polylines, record their lengths, and prompt for the length of one specified by color.

**Variants.** $K \in \{2, \ldots, 5\}$ with tiling, palette, and length targets varying by seed.

**Difficulty controls.** The correct line length is the measure of difficulty.

**Answer type.** Integer length.

### B.2.4 TILES LINE INTERSECTIONS

**Problem.** Intersection counting over colored polylines constrained to tile edges.

**Construction.** Build a vertex graph for the selected tiling; lay out $k$ vertex-simple polylines with distinct colors and no shared edges.

**Variants.** $k \in \{2, \ldots, 5\}$ with tiling family and target count sampled per instance.

**Difficulty controls.** The number of intersections measures difficulty.

**Answer type.** Integer number of shared vertices (including endpoints).

### B.2.5 TILES RECOLORING

**Problem.** Cell-wise recoloring/difference counting between two related boards.

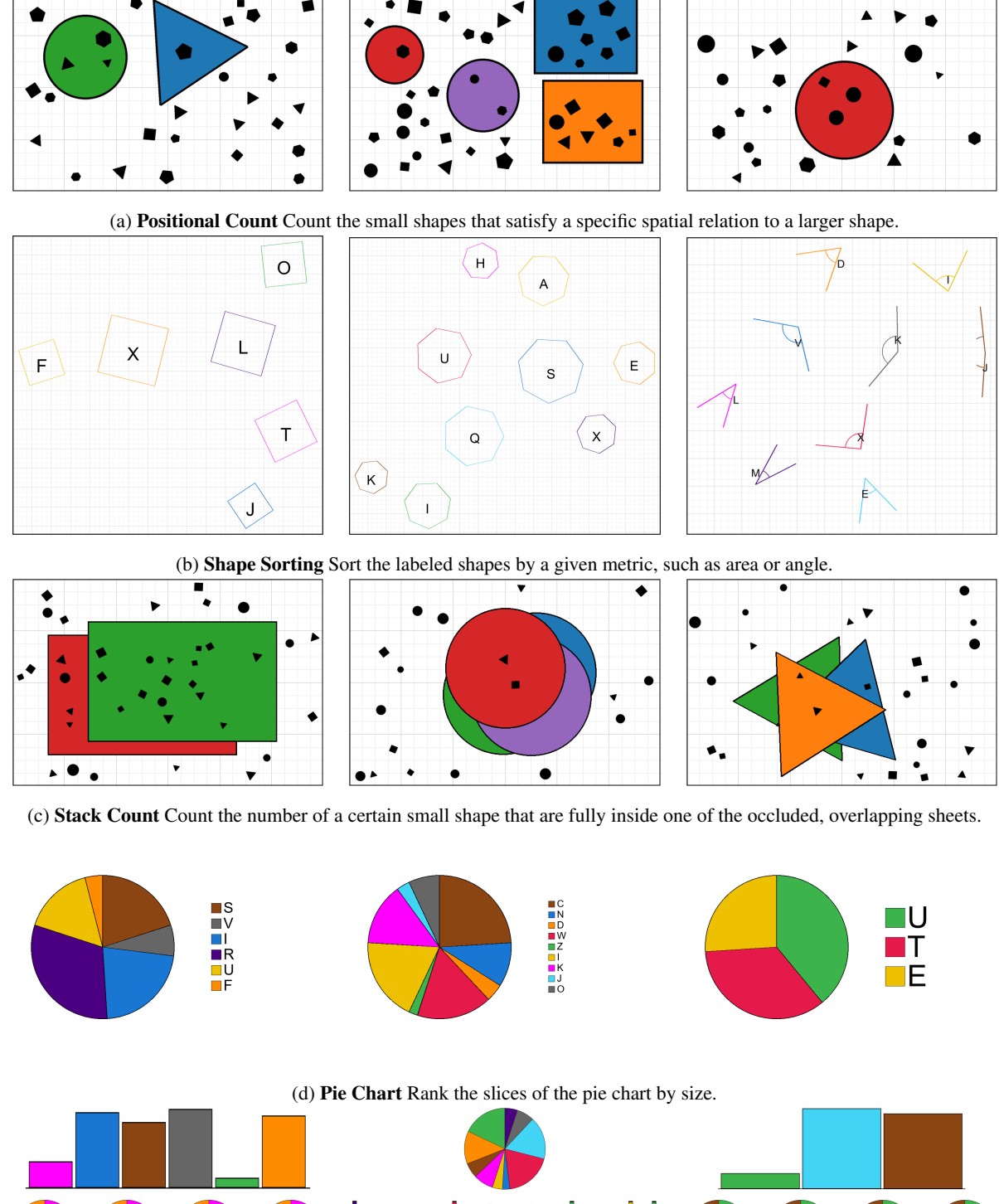

(a) **Positional Count** Count the small shapes that satisfy a specific spatial relation to a larger shape.

(b) **Shape Sorting** Sort the labeled shapes by a given metric, such as area or angle.

(c) **Stack Count** Count the number of a certain small shape that are fully inside one of the occluded, overlapping sheets.

(d) **Pie Chart** Rank the slices of the pie chart by size.

(e) **Chart Comparison** Find the bar/pie chart that correctly represents the proportions in the top chart.

Figure 12: Examples of Geometric Reasoning and Chart tasks.

**Construction.** Grow a connected region on the left board; derive the right board by adding/removing a connected set (same-color variant) or additionally recoloring overlap (color-change variant).

**Variants.** Two variants - same color vs. color change - across several tiling families.

**Difficulty controls.** The number of different cells measures the difficulty.

**Answer type.** Integer number of differing cells.

## B.3 SYMMETRY & PATTERN RECOGNITION

Figure 14 shows examples of the this type of task.

### B.3.1 MIRROR IDENTIFICATION

**Problem.** Textual classification of mirror symmetry (including "none") for a composite scene.

**Construction.** Place motif instances inside class-specific fundamental regions to synthesize scenes. Verify the final bitmap's category via color-aware symmetry tests; pair with six textual options and shuffle.

**Variants.** Six labels - vertical, horizontal, main diagonal, anti-diagonal, vertical+horizontal, none - with target count and canvas scale adapted to the class.

**Distractors.** The five incorrect textual descriptions serve as distractors; all six labels are offered.

### B.3.2 SYMMETRY FILL

**Problem.** Grid completion under a specified mirror constraint. A 2×2 grid is shown with one tile missing; select the tile that restores the target symmetry.

**Construction.** Render a base tile, apply the rule (vertical, horizontal, both, main-diagonal, anti-diagonal) to fill the grid, remove one tile, and construct options by applying distinct transforms while enforcing pairwise distinctness.

**Variants.** Five rule keys as above; missing position and motif vary.

**Distractors.** Transform pool filtered to retain only visually distinct candidates; select three and shuffle with the correct transform.

### B.3.3 FRIEZE GROUPS

**Problem.** Odd-one-out identification among four horizontal strips, each generated from a frieze symmetry; three share the same neighbor rule, one differs.

**Construction.** Sample a motif family; choose a majority frieze group for three strips and a distinct group for the odd strip. Render with consistent spacing and label (a-d).

**Variants.** Six Conway frieze groups (step, sidle, jump, spinning hop, spinning sidle, spinning jump). Strip length and option order vary per instance.

**Distractors.** The distractors are simply additional strips from the majority frieze class; the odd class is unique by construction.

### B.3.4 WALLPAPER GROUPS

**Problem.** Odd-one-out among four 2D wallpaper patches; three are sampled from one wallpaper group and one from another.

**Construction.** Sample a motif family and wallpaper groups; generate patches under each group, crop to equal square tiles, and compose a labeled 2×2 grid.

**Variants.** Seventeen IUC wallpaper groups; majority/odd selection and option order are randomized.

**Distractors.** The three majority-group patches form the distractor set by construction.

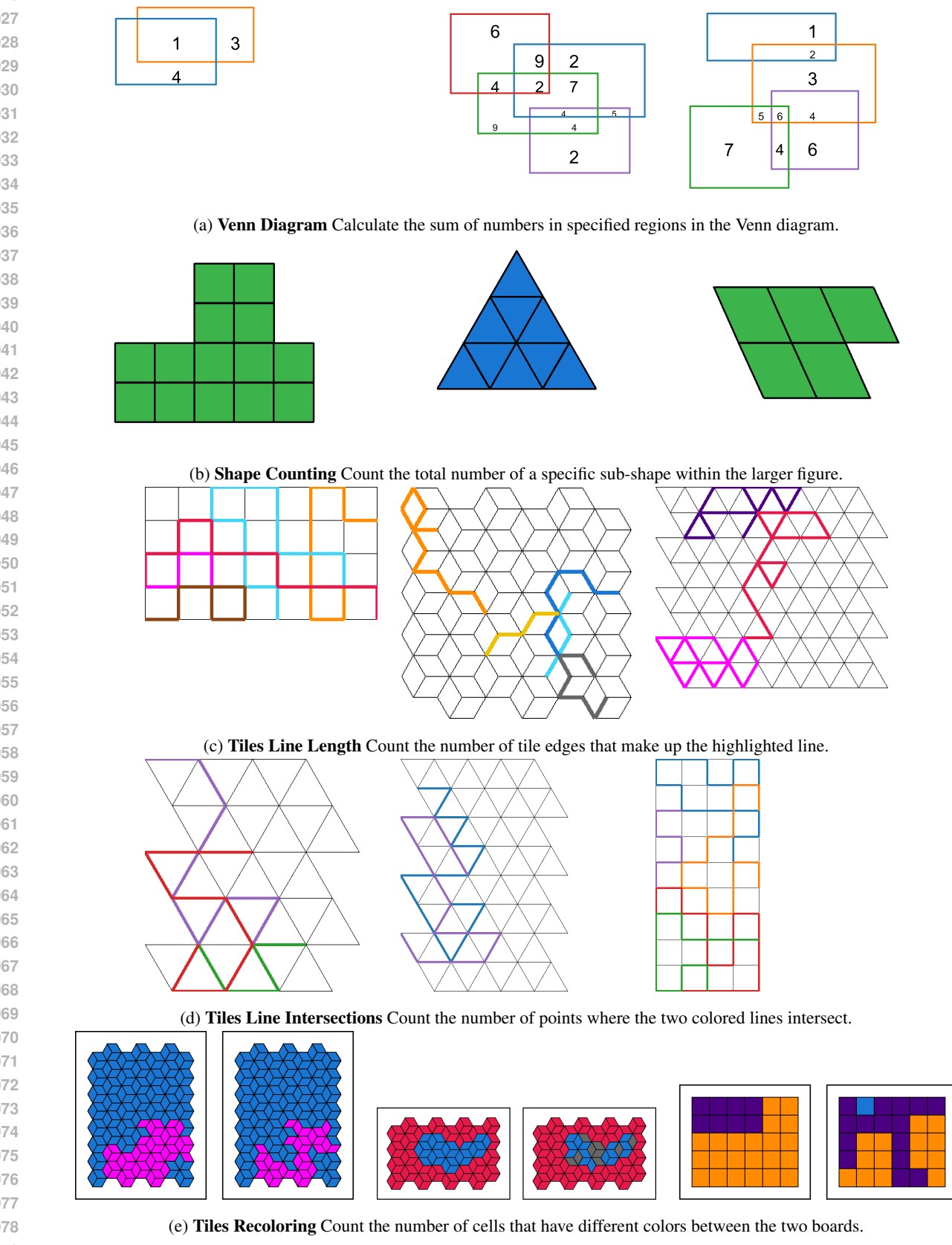

(a) **Venn Diagram** Calculate the sum of numbers in specified regions in the Venn diagram.

(b) **Shape Counting** Count the total number of a specific sub-shape within the larger figure.

(c) **Tiles Line Length** Count the number of tile edges that make up the highlighted line.

(d) **Tiles Line Intersections** Count the number of points where the two colored lines intersect.

(e) **Tiles Recoloring** Count the number of cells that have different colors between the two boards.

Figure 13: Examples of Counting tasks.

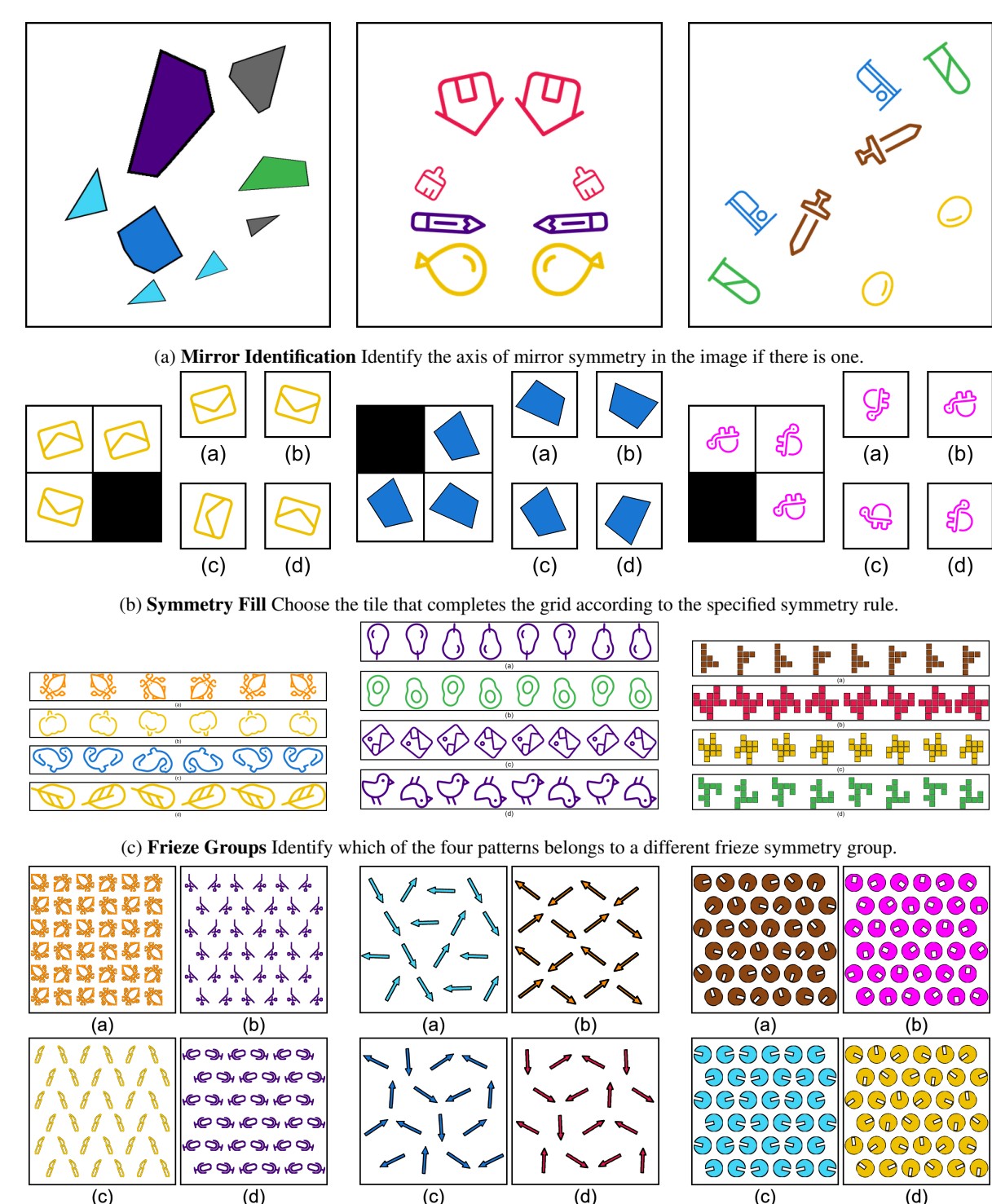

(a) **Mirror Identification** Identify the axis of mirror symmetry in the image if there is one.

(b) **Symmetry Fill** Choose the tile that completes the grid according to the specified symmetry rule.

(c) **Frieze Groups** Identify which of the four patterns belongs to a different frieze symmetry group.

(d) **Wallpaper Groups** Identify which of the four patterns belongs to a different wallpaper symmetry group.

Figure 14: Examples of Symmetry tasks.

### B.4 Sequence & Transformation Reasoning

Figure 15 shows examples of the this type of task.

#### B.4.1 Transform Result Identify

**Problem.** Visual selection of the result of applying a sampled transform to the original tile.

**Construction.** Render a motif patch, center it on graph paper, sample a transform from TF_RULES, and construct one correct and three incorrect image options with consistent placement and borders. Compose a top/bottom layout with labels.

**Variants.** Eight transform families; translations use randomized vectors.

**Distractors.** Render alternative transforms (including alternative translation vectors) and retain only candidates that are pairwise distinct.

#### B.4.2 Transform Pair Infer

**Problem.** Identify the single transformation that maps a source tile to a target tile; "none of the above" may be correct by omission.

**Construction.** Render an asymmetrized motif on graph paper, choose a true transform from mirrors/rotations/translation, synthesize the target, and verify uniqueness against the full rule set. Compose a side-by-side display with an arrow and six labeled textual options.

**Variants.** Up to eight answer classes: seven concrete transforms (vertical mirror, horizontal mirror, main diagonal mirror, anti-diagonal mirror, 90° rotation, 180° rotation, 270° rotation, translation) plus none (correct with probability 1/6 when the true transform is withheld).

**Distractors.** When the true transform is present, sample other transforms as distractors with uniqueness filtering; when omitted, append none and select the remainder accordingly (with none fixed to the final slot for clarity).

#### B.4.3 Transform Similarity Identify

**Problem.** Similarity-based selection under Euclidean similarity (uniform scale + $D_4$ rigid/mirror motions). Either select the single similar option, or the single dissimilar one.

**Construction.** Render an asymmetrical motif and produce options via allowed $D_4$ transformations with optional uniform scaling and translation. For "dissimilar", apply enabled breaker warps (e.g., anisotropic scale, shear, perspective) and reject near-similar outcomes via a canonical checker.

**Variants.** Two core variants (one similar, one dissimilar) with four options.

**Distractors.** For "similar", distractors are other (dis)allowed outcomes that remain distinct; for "dissimilar", distractors are similar options.

#### B.4.4 Sequence Rotation

**Problem.** Rotation-only progression over a single bitmap with a constant angular step; one panel is masked.

**Construction.** Render a base motif, compute a global scale fitting all sampled rotations, and generate tiles using a step from $\{30°, 45°, 60°, 90°\}$ in either direction. Mask one panel and present four options.

**Variants.** Eight rotation regimes (four step sizes × two directions); mask index is uniform.

**Distractors.** Alternative rotation angles filtered by separation thresholds; weakly separated candidates are rejected.

#### B.4.5 Sequence Arithmetic

**Problem.** Next-step prediction in a count-based progression with one masked panel.

**Construction.** Sample a motif by weights. Draw a sequence with the count changing by a set increment/decrement; mask one panel and provide four choices.

**Variants.** Two architectural paths (direct-count vs. repeated-layout), four layout templates, and a uniformly sampled mask index.

**Distractors.** Different incorrect counts are made and checked for enough visual difference from other options.

### B.4.6 SEQUENCE MULTI-COLUMN ARITHMETIC

**Problem.** Multi-column next-step prediction where each column follows its own arithmetic progression.

**Construction.** Sample 2-6 columns, motif kinds, and per-column base specs; draw four time steps using a shared within-column scale set by the maximum count. Hide the final panel and provide four candidates for the continuation.

**Variants.** Continuous parameterization over column count, motifs, and steps; the core schematic is fixed (three observed panels, one to predict).

**Difficulty controls.** The number of columns is used to measure difficulty.

**Distractors.** Edit exactly one column per wrong option, escalating $\pm\Delta$ until the local change exceeds a threshold; reject duplicate/low-contrast candidates.

### B.5 TOPOLOGICAL & GRAPH REASONING

Figure 16 shows examples of the this type of task.

### B.5.1 TILES GEOMETRY

**Problem.** Geometric measurement over colored regions on a tiling (area, perimeter, holes, area difference, union perimeter).

**Construction.** Sample a tiling, paint disjoint regions, compute region graphs, and evaluate the requested measure. Render a crisp board on white with a natural-language prompt.

**Variants.** Five query types - single region area, single region perimeter, single region hole, two region area difference, union of two region perimeter - with per-instance color selection.

**Difficulty controls.** The size of the tiling is the measure of difficulty.

**Answer type.** Integer.

### B.5.2 TILES CONNECTED COMPONENT

**Problem.** Component analysis on a colored tiling. Query the size of the largest/smallest component or the number of components within a specified color under a given adjacency notion.

**Construction.** Sample a tiling and a non-uniform coloring; build the dual graph with edge adjacency (or point-touch for circular tilings). Compute per-color connected components and select a query with a unique answer (enforced for extreme queries).

**Variants.** Six combinations from three measures (largest size, smallest size, count components) $\times$ two adjacency regimes (edge vs. point-touch when applicable).

**Difficulty controls.** The number of components measures difficulty.

**Answer type.** Integer.

### B.5.3 TILES SHORTEST PATH

**Problem.** Shortest-path computation on a cell graph with obstacles; return the minimum number of edge-steps or -1 if unreachable.

**Construction.** Sample a tiling, build the dual graph, sample an obstacle field from beta-regime priors (sparse, dense, balanced, patchy), choose start/end tiles, and use BFS to verify distance or enforce unreachable cases.

**Variants.** There are four obstacle regimes. With probability 0.1, unreachable instances are generated.

**Difficulty controls.** The length of the shortest path is the difficulty.

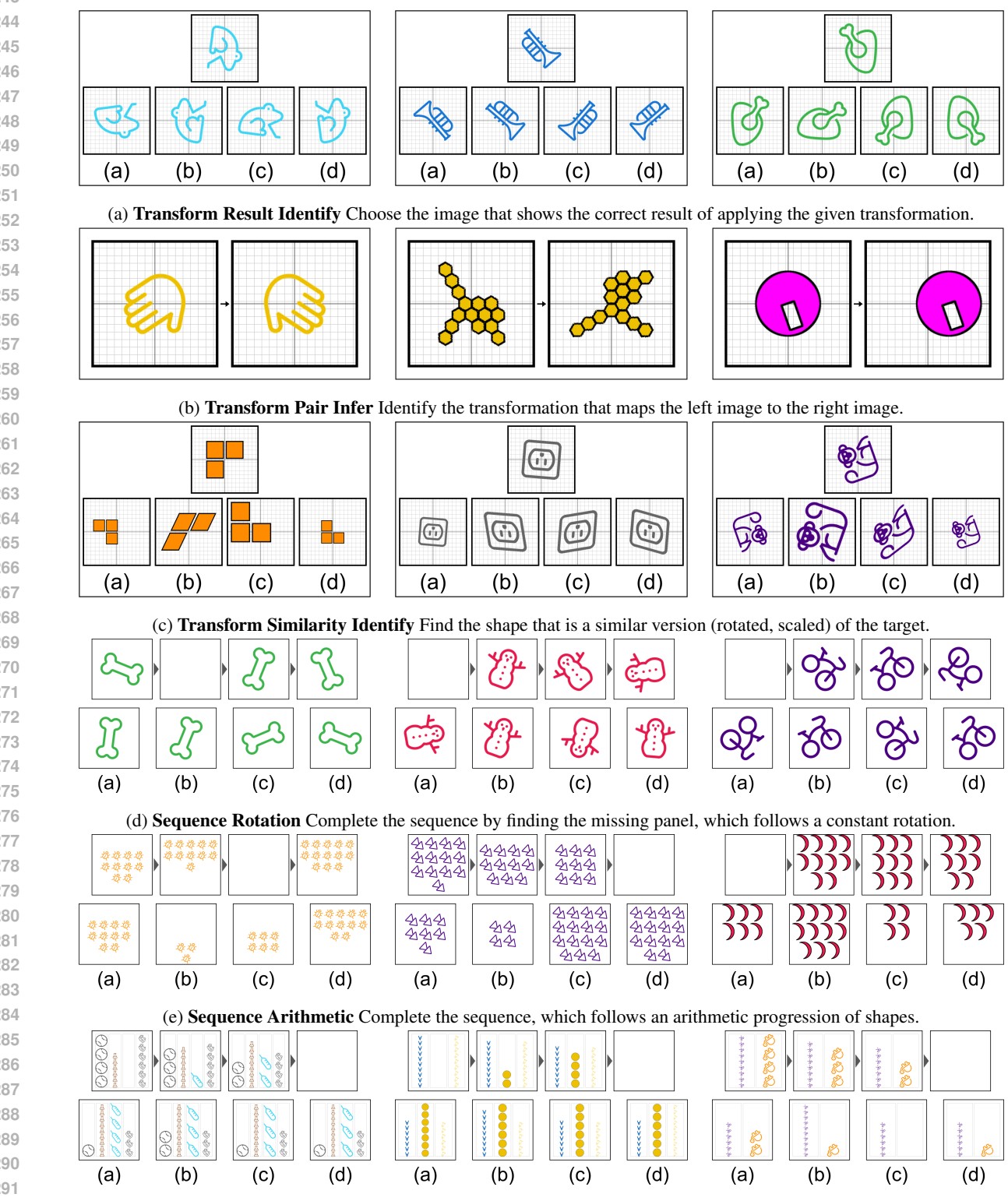

(a) **Transform Result Identify** Choose the image that shows the correct result of applying the given transformation.

(b) **Transform Pair Infer** Identify the transformation that maps the left image to the right image.

(c) **Transform Similarity Identify** Find the shape that is a similar version (rotated, scaled) of the target.

(d) **Sequence Rotation** Complete the sequence by finding the missing panel, which follows a constant rotation.

(e) **Sequence Arithmetic** Complete the sequence, which follows an arithmetic progression of shapes.

(f) **Sequence Multi-Column Arithmetic** Predict the final panel where each column follows its own progression.

Figure 15: Examples of Transformation and Sequence tasks.

**Answer type.** Integer (distance) or -1.

### B.5.4 MISSING TILES

**Problem.** Completion of a partially blanked tiling via color restoration or shape fitting (orientation changes allowed).

**Construction.** Sample a tiling and remove a connected region of bounded size. In the color variant, recover the exact color assignment for the missing cells. In the shape variant, recover the exact shape up to the tiling's dihedral symmetries.

**Variants.** Two balanced variants (color vs. shape) across four tilings (square, triangular, hexagonal, rhombille).

**Difficulty controls.** The size of the tiling is used to measure difficulty

**Distractors.** Color variant performs pairwise color swaps or Dirichlet-weighted palette shuffles; shape variant samples alternative connected subsets of equal size that are non-congruent under allowed symmetries.

### B.5.5 TILES COMPOSITION

**Problem.** Piece equivalence and assembly. Either decompose a connected region into a multiset of connected pieces (bags) or compose a bag into a single connected target.

**Construction.** Sample a tiling and connected region; split into 2-5 connected pieces via randomized BFS growth. In "decompose", show the region on top and candidate bags below; in "compose", show a bag on top and candidate target shapes below. Normalize framing across options.

**Variants.** Two modes (decompose vs. compose) × two color modes (uniform, random_per_cell). Additional variation from piece counts and tiling families.

**Difficulty controls.** The number of connected pieces is used as a measure for difficulty.

**Distractors.** For decompose, bags reuse piece cardinalities but alter piece shapes. For compose, candidates match area but do not correspond to the true union of pieces.

## C HUMAN EVALUATION

We conducted a controlled human evaluation using a custom-built web application. Participants accessed the app through a browser and were assigned a of 25 problems (or 10 problems if explicitly chosen by the participant). Each problem consisted of a visual prompt (image and/or text) and an input field for responses.

This setup allowed us to systematically measure accuracy, timing, and subjective feedback across participants and tasks, enabling comparison of human performance against large language models (LLMs).

The application enforced basic validation (e.g., number formats, single-choice letters, or ordered lists) to ensure responses were well-formed. For each participant, we recorded:

- Response text
- Correctness (with respect to the ground truth)
- Per-question time taken
- Overall completion time
- Types of tasks assigned

To reduce variability in prior knowledge, the interface also provided a dedicated *Definitions* panel containing concise explanations of key terms and concepts (e.g., symmetry, rotation, translation). This feature ensured that all participants could engage with the tasks from a comparable baseline of conceptual understanding, thereby minimizing confounds due to varying background knowledge.

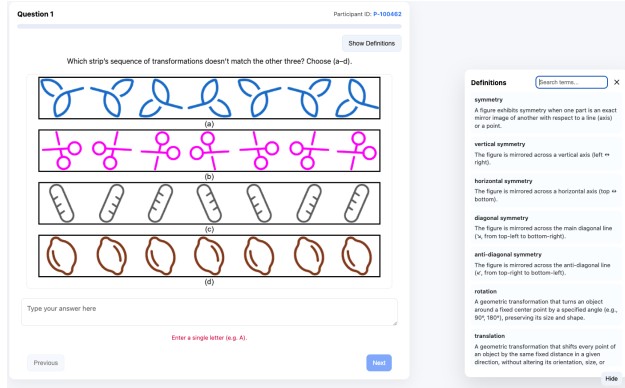

Figure 17: Web application interface used for the human evaluation. Participants were shown a visual prompt (image and/or text) and provided responses in the answer box.

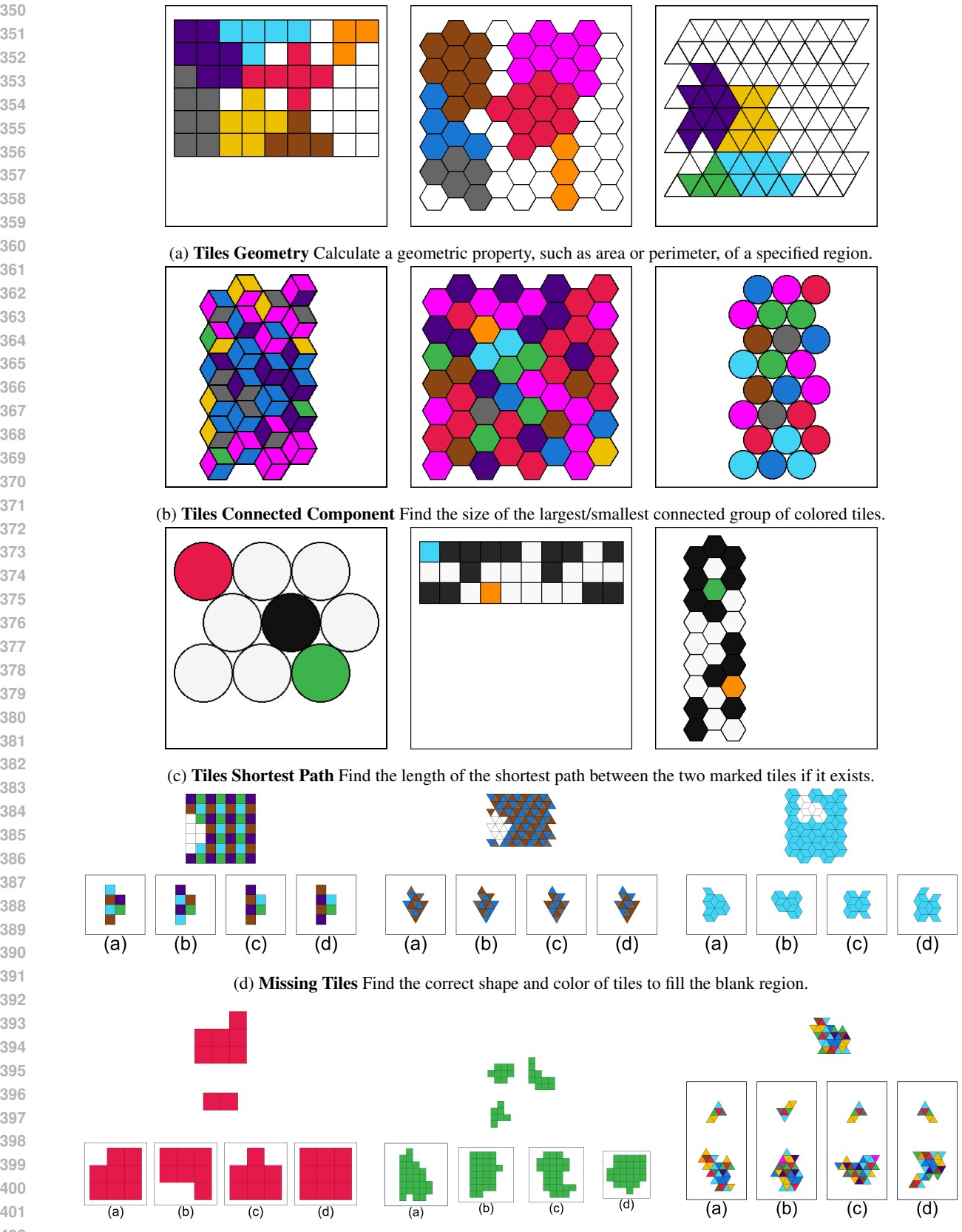

(a) **Tiles Geometry** Calculate a geometric property, such as area or perimeter, of a specified region.

(b) **Tiles Connected Component** Find the size of the largest/smallest connected group of colored tiles.

(c) **Tiles Shortest Path** Find the length of the shortest path between the two marked tiles if it exists.

(a)    (b)    (c)    (d)    (a)    (b)    (c)    (d)    (a)    (b)    (c)    (d)

(d) **Missing Tiles** Find the correct shape and color of tiles to fill the blank region.

(a)    (b)    (c)    (d)    (a)    (b)    (c)    (d)    (a)    (b)    (c)    (d)

(e) **Tiles Composition** Choose the option that composes or decomposes into the top figure.

Figure 16: Examples of Topological and Tiling tasks.

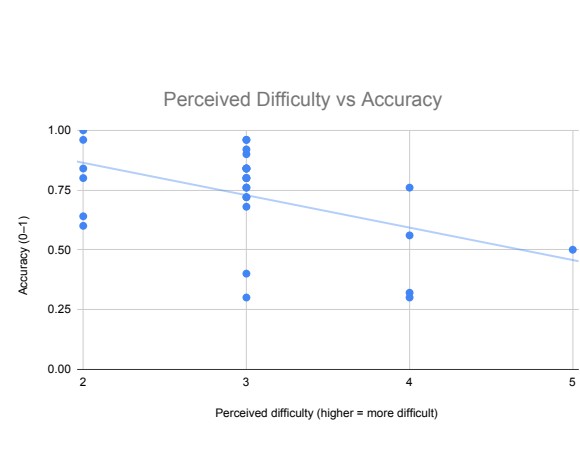

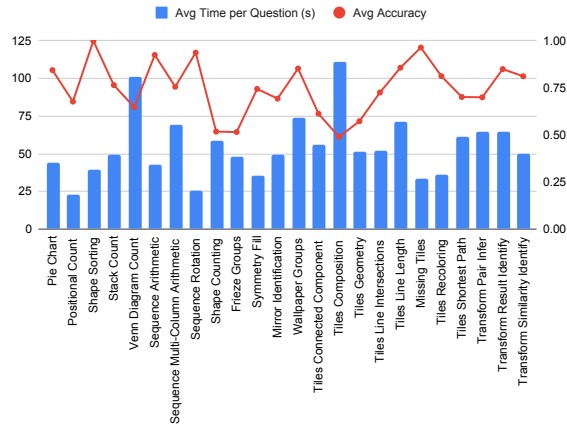

(a) Perceived difficulty vs accuracy.

(b) Average time per question (bars) and average accuracy (line) per task.

Figure 18: Human evaluation results. (a) Scatter plot of participant perceived difficulty versus accuracy (b) Task-level time and accuracy.

After completing the problem set, participants filled out a *post-questionnaire survey* in which they rated the perceived difficulty, clarity, familiarity, and engagement, along with providing optional feedback.

## C.1 HUMAN EVALUATION SETUP

Figure 17 shows the web interface used for collecting human responses for the assigned tasks implemented specifically for SPHINX.

## C.2 HUMAN PERFORMANCE ANALYSIS

Figure 18 shows the human performance on the evaluation tasks, highlighting accuracy distributions, time–accuracy analysis, and the relationship between subjective difficulty ratings and objective outcomes.

