# OpenReview forum: "Sphinx: Visual Perception and Reasoning Gym"
_ICLR.cc/2026/Conference — ICLR 2026 Conference Withdrawn Submission_

### Official Review · Reviewer_SvJC · 2025-10-24

**Soundness:** 2
**Presentation:** 3
**Contribution:** 2
**Rating:** 2
**Confidence:** 4

**Summary:**

This paper introduces SPHINX, a synthetic benchmark designed to evaluate visual perception and reasoning abilities of multimodal large language models (MLLMs). SPHINX programmatically generates tasks based on three composable modules — motifs, tilings, and tasks — spanning 25 visual reasoning types such as symmetry detection, geometric transformations, and spatial or topological reasoning.

**Strengths:**

1. The factorized “motif–tiling–task” architecture allows systematic control over appearance, geometry, and rule complexity. This modularity is elegant and facilitates extensibility.
2. The implementation is detailed, reproducible, and bridges cognitive-style reasoning benchmarks (e.g., ARC, Raven) with multimodal settings.

**Weaknesses:**

1. The main contribution is a benchmark and accompanying analysis; there is no novel learning mechanism, training paradigm, or model design. The RLVR section applies existing GRPO and EasyR1 frameworks with minimal modification.
2. The claimed “synthetic reasoning gym with verifiable rewards” closely parallels Reasoning Gym (Stojanovski et al., 2025) and Enigmata (Chen et al., 2025). SPHINX’s focus on visual primitives is interesting but not fundamentally new.
3. The reinforcement learning experiments confirm that verifiable rewards can improve performance, but lack deeper analysis (e.g., ablations, reward shaping, generalization mechanisms). The gains seem mechanical rather than conceptually informative.

**Questions:**

In addition to the weaknesses mentioned above, I have several specific questions and concerns:
1. The paper claims to define 25 tasks across five categories, but Appendix A.2 states that only 12 tasks are currently implemented. Could the authors clarify whether the remaining tasks are conceptual, partially implemented, or part of future work?
2. The current contribution feels more like a well-engineered and systematic consolidation of existing reasoning tasks rather than a new conceptual advance. How does SPHINX fundamentally change our understanding or evaluation of multimodal reasoning, beyond integrating known task types under one unified framework?
3. While I appreciate the engineering quality and thorough experimentation, the methodological component (particularly the RLVR section) appears quite incremental. It would be helpful if the authors could provide clearer evidence of methodological novelty or broader empirical impact.

---

### Official Review · Reviewer_C3wo · 2025-10-29

**Soundness:** 3
**Presentation:** 4
**Contribution:** 3
**Rating:** 6
**Confidence:** 4

**Summary:**

The paper presents SPHINX, a synthetic gym for evaluating visual perception and reasoning in multimodal models. It introduces 25 procedurally generated, verifiable tasks that span geometric, spatial, and sequential reasoning, providing a systematic framework for scalable training and assessment. Results show large performance gaps between current models and humans, while reinforcement learning with verifiable rewards improves both accuracy and generalization.

**Strengths:**

The paper has the following strengths:
- Comprehensive benchmark coverage: SPHINX unifies low-level perception and high-level reasoning tasks under a single verifiable framework.

- Methodological clarity: The procedural generation pipeline and RLVR integration make the evaluation reproducible and interpretable.

**Weaknesses:**

I found the following weaknesses in the paper:
- The benchmark remains limited to synthetic, well-defined tasks. Broader multimodal reasoning in real-world or noisy contexts is not yet captured, and the multiple-choice setup may inflate apparent gains through guessability rather than genuine reasoning.
- The benchmark consists mainly of general MLLMs and lacks domain-specific MLLMs. Their performance on these geometric tasks can be benchmarked, and how they can be improved using the RL framework.

**Questions:**

I have the following suggestions:
- Some more domain-specific MLLMs, such as G-LLaVA, Math-LLaVA, Math-Puma, and a general MLLM - Janus Pro can be included in the paper

---

### Official Review · Reviewer_kRCV · 2025-10-31

**Soundness:** 3
**Presentation:** 2
**Contribution:** 2
**Rating:** 4
**Confidence:** 3

**Summary:**

The paper introduces SPHINX, a synthetic gym for generating and evaluating visual perception and reasoning tasks. Its core innovation is a modular, procedurally generated framework that decouples visual motifs, spatial tilings, and reasoning tasks, enabling the creation of a vast dataset with unambiguous, verifiable ground-truth solutions.

**Strengths:**

1.The paper is well written and easy to follow
2.The proposed benchmark I think is solid, and the question in the benchmark is hard enough to challenge current MLLMs
3.The authors conducted extensive experiments to evaluate the limitations of current vision-language models.

**Weaknesses:**

1.The experimental results presented in this paper, which utilize SPHINX data for reinforcement learning, are not entirely convincing. The performance improvements observed on benchmarks like MM-IQ and Geo3K may stem from the similarity between SPHINX-generated tasks and those in the benchmarks, rather than reflecting genuine advances in general reasoning capability. Consequently, the model's performance on more general-purpose benchmarks may remain limited. This could ultimately relegate SPHINX to the status of a specialized evaluation dataset similar to MM-IQ, rather than a driver of broad progress. Therefore, I would like to see more substantial improvements demonstrated on universal visual understanding benchmarks.
2.I am concerned that the improvements achieved on SPHINX's controlled, synthetic tasks may not effectively transfer to solving real-world visual reasoning problems. The latter typically involve noisy natural images and more complex, open-ended semantics. This suggests that while the benchmark is well-suited for diagnosing specific perceptual primitives, it might be too narrow to drive meaningful progress in general multimodal reasoning.
3.The paper lacks a detailed description of the human evaluation methodology. Having attempted some sample tasks myself and found them almost entirely solvable（lol）, I would like the authors to provide a comprehensive explanation of how the human accuracy rates were collected and validated.

**Questions:**

See Weakness.

---

### Official Review · Reviewer_aCVS · 2025-11-01

**Soundness:** 3
**Presentation:** 2
**Contribution:** 3
**Rating:** 4
**Confidence:** 3

**Summary:**

This paper presents SPHINX, a large-scale synthetic gym for evaluating and improving visual perception and reasoning for MLLMs. SPHINX procedurally generates diverse, verifiable problems—covering 25 task types across geometry, counting, symmetry, sequence, and topology—each paired with ground-truth solutions for precise evaluation.

Extensive benchmarking shows that even GPT-5 attains only 47.3% accuracy, far below the human baseline (75.4%), while open-source MLLMs such as Qwen2.5-VL-32B reach just 32.2%. Tasks involving transformations, symmetry, and tile-based reasoning remain most difficult.

The authors further apply Reinforcement Learning with Verifiable Rewards (RLVR) on Qwen2.5-VL models, which improves accuracy and generalization, demonstrating that structured, verifiable synthetic environments like SPHINX can effectively advance multimodal understanding and reasoning capabilities.

**Strengths:**

1. Comprehensive synthetic benchmark for MLLM perception and reasoning. SPHINX systematically evaluates multimodal models across 25 task types—including geometric, counting, symmetry, sequence, and topological reasoning. Each instance is paired with verifiable ground truth, allowing precise and scalable assessment. This dataset targets core visual–cognitive primitives rather than high-level captioning or description tasks, providing both diagnostic and training value.
2. Demonstrated effectiveness of RLVR on SPHINX for generalization. The integration of RLVR on SPHINX-generated data leads to notable performance gains. Specifically, Qwen2.5-7B improves from 25.2% to 42.6% on in-distribution tasks and shows consistent improvements on out-of-distribution and external benchmarks (e.g., MathVision, MM-IQ, Geo3k), providing empirical evidence that verifiable synthetic supervision can enhance general reasoning generalization, establishing SPHINX as a valuable environment for multimodal RL research.

**Weaknesses:**

1. Limited depth in analysis. The discussion (Section 4.2) focuses mainly on pairwise accuracy differences (e.g., GPT-5 vs. GPT-5-Mini) without identifying specific cognitive failure types. Furthermore, the analysis on reasoning vs. non-reasoning behaviors is missing but important. Since human evaluators only achieve 75.4% accuracy, far from perfect performance, it would also be informative to analyze human error patterns. Such analysis could reveal whether remaining errors arise from ambiguous task construction or genuine difficulty, thereby further demonstrating SPHINX’s reliability as a benchmark.
2. Evaluation breadth and diagnostic clarity on RLVR models. The current evaluation on two RLVR models primarily covers the SPHINX dataset and three additional specialized benchmarks (MathVision, MM-IQ, Geo3K. However, it lacks assessment on more general multimodal benchmarks (e.g., MMBench, MMMU, SEED-Bench), which would clarify whether SPHINX-based RL affects broader visual–language capability. Moreover, even after RLVR fine-tuning, the in-domain accuracy remains below 50%, suggesting potential issues with reward sparsity by over-complex task generation (always getting 0 reward). The authors could further investigate whether models receive insufficient reward signals for difficult samples or explore distillation (e.g., from GPT-5) for cold-start to address sparse supervision. At least a deeper discussion of these possibilities would make the experimental conclusions more convincing and actionable.

**Questions:**

1. It is helpful to provide some insights about why human performance is only 75%. Otherwise the reliability of the dataset is questionable.
2. Providing the RLVR model’s performance on more general multimodal benchmarks could further enhance the paper’s contribution.
3. Seems that the paper is wider than standard ICLR template, even if the page count (9) is technically within limit, changing the width (thus fitting more content per page) undermines the spirit of the page-limit rule. The authors should revert to the official template dimensions to ensure fairness.

---

### Note · Authors · 2025-11-12

I have read and agree with the venue's withdrawal policy on behalf of myself and my co-authors.